# Prevalence, Risk Factors, and Multimorbidity Patterns in Climacteric Women with Hypertension

**DOI:** 10.3390/ijerph22091360

**Published:** 2025-08-29

**Authors:** Juliene Gonçalves Costa, Ana Luiza Amaral, Julia Buiatte Tavares, Aline Keli de Oliveira, Ana Clara Ribeiro Cunha, Juliana Cristina Silva, Guilherme Morais Puga

**Affiliations:** 1School of Human Sciences (Exercise and Sport Science), The University of Western Australia, Perth 6009, Australia; juliene.goncalvescostadechichi@research.uwa.edu.au; 2Laboratory of Cardiorespiratory and Metabolic Physiology, Physical Education and Physical Therapy Department, Federal University of Uberlândia, Uberlândia 38400-678, Brazil; ana.amaral@ufu.br (A.L.A.); juliabuiatte@ufu.br (J.B.T.); anaclararibeirocunha1@gmail.com (A.C.R.C.); julianasilvacristina@yahoo.com.br (J.C.S.); 3Medical Clinic, Clinical Hospital of Federal University of Uberlândia, Pará Avenue, Uberlândia 38405-320, Brazil; alinekeli@ymail.com

**Keywords:** menopause, chronic non-communicable diseases, aging, high blood pressure

## Abstract

Although the relationship between risk factors and disease patterns still remains poorly understood, arterial hypertension in climacteric women is a substantial risk factor for multimorbidity. This cross-sectional study analyzed data from 1003 women aged ≥40 years attending Brazilian Basic Health Units to assess multimorbidity (≥2 chronic conditions) and its patterns (cardiometabolic, musculoskeletal, and neuropsychological). An adjusted logistic regression revealed that postmenopausal status (OR: 2.17; 95% CI: 1.05–4.48) and an age of ≥70 years (OR: 2.85; 95% CI: 1.16–6.99) were key risk factors for multimorbidity. Notably, 86% of hypertensive women had multimorbidity, most frequently dyslipidemia (50%), type 2 diabetes (37%), and thyroid disorders (18%). The cardiometabolic pattern (86% prevalence) was strongly associated with hypertension, especially among women aged ≥50 years (OR: 2.10; 95% CI: 1.10–3.98) and those with obesity grade I+ (OR: 2.30; 95% CI: 1.36–3.89). Musculoskeletal disorders were associated with postmenopausal status (OR: 2.41; 95% CI: 1.05–5.51) and obesity (OR: 1.92; 95% CI: 1.08–3.43), while neuropsychological diseases showed no significant associations. These findings highlight that hypertensive climacteric women—especially postmenopausal, older, or those with obesity—face elevated risks of cardiometabolic and musculoskeletal multimorbidity, underscoring the need for targeted preventive strategies in this population.

## 1. Introduction

Multimorbidity is defined, according to the WHO, as the co-occurrence of two or more chronic medical conditions in a person and includes physical and mental health conditions [1]. The coexistence of multiple chronic diseases is a growing problem and a global challenge for health systems, with complexity for both health professionals and researchers [2].

People with multiple chronic diseases have total costs 5.5 times higher and each additional chronic condition was associated with an increase of 3.2 visits and 33% in costs. In addition, they have a greater number of medical appointments, with an annual average of 15.7 compared to 4.4 in the non-multimorbid sample, and also a greater use of medications, polypharmacy being quite common among these patients [3,4].

In Brazil, multimorbidity affects one in every five adults with two or more chronic diseases and one in every ten has more than three diseases, which represents more than 43 million and 20 million Brazilians, respectively. The prevalence varies among the regions of the country, with the south and southeast regions having the highest rates. The states of Santa Catarina and Rio Grande do Sul have a prevalence of 26% to 29% for >2 diseases, followed by Paraná, São Paulo, and Rio de Janeiro with 23% to 25% [5].

Of the diseases reported, arterial hypertension (HT) (22.3%), spinal problems (19%), and dyslipidemia (8.4%) were the most prevalent diseases, with different patterns between men and women [5]. In the study by Araújo et al. [6], the factor with the greatest strength of association in women was heart disease, while in men, an association was identified in two groups, and lung disease was the disease with a higher factor loading.

Although blood pressure (BP) levels increase with age in both sexes, the increase per decade is greater in women (8.1 mmHg) than in men (4.7 mmHg) [7]. Due to aging, hypoestrogenism, and lifestyle factors, climacteric women are increasingly presenting multiple chronic diseases [8]. Furthermore, variations in gender roles have a significant impact on health outcomes. Women are often more inclined to assume the primary caregiver role, dedicating more time to caregiving, and this often results in poorer self-reported health among female caregivers compared to their male counterparts, who typically spend less time on caregiving tasks. These patterns persist into later life, leading to greater physical and mental burdens as they manage their own health issues along with caregiving responsibilities [9].

Women exhibited higher rates of multimorbidity and polypharmacy compared to men, as well as a greater burden of disease and stress symptoms [10]. Women have a different prevalence, risk factors, and disease pattern than men, and studies that analyze this population in greater depth are extremely important. HT is the most frequent chronic disease in this population; therefore, in addition to understanding the prevalence and types of diseases, it is necessary to understand the most frequent associations and patterns, as well as the specific risk factors associated with the most prevalent diseases, such as HT.

While multimorbidity in aging populations is extensively documented [11,12], the understanding of its specific patterns and risk factors in vulnerable demographic groups, such as climacteric women with hypertension, remains limited, particularly within the context of Brazilian public health [6,13,14]. The majority of studies focus on the overall prevalence of multimorbidity or the association of individual diseases without exploring the specific combinations of chronic conditions that coexist in this population [15]. Furthermore, the menopausal transition represents a critical period of hormonal and physiological changes that exacerbate the risk of certain diseases, demanding a more in-depth investigation into how these factors intertwine [14,16,17].

Therefore, this study seeks to fill this gap by analyzing, in the menopausal population, the patterns of multimorbidity (cardiometabolic, musculoskeletal, and neuropsychological) and identifying the specific risk factors (age, menopausal status and body mass index (BMI)) associated with each pattern in women with hypertension followed by basic health units (UBS). By focusing on the combinations of diseases and their determinants, we offer a perspective that goes beyond a simple count of comorbidities, providing more detailed insights into the development of public health strategies and clinical interventions targeted at this vulnerable population. Therefore, the aim of this study is to recognize the associated disease patterns and the specific risk factors for each pattern.

## 2. Materials and Methods

### 2.1. Study Design

This is a cross-sectional study of a sample of the urban female population over 40 years old registered in the Hyperdia Program and undergoing medical follow-up at UBSs in the city of Uberlândia, Minas Gerais, Brazil. UBSs are primary healthcare centers where Family Health teams carry out a range of health actions. They represent the main entry point to the Brazilian Unified Health System (SUS), addressing individual and collective health needs. UBSs serve the general population, offering primary healthcare, including prevention, diagnosis, and treatment. They are an integral part of the SUS, regulated by federal laws and regulations that guide their structure and operation [18]. The data for this study was collected from a system between 11 October and 22 December 2021, from an analysis of the medical records of participants in the Hyperdia Program, a SUS program that monitors people with hypertension and/or diabetes. The program serves as a primary care tool, providing continuous follow-up and monitoring of blood pressure and glucose levels for these patients. The study was conducted by the Laboratory of Cardiorespiratory and Metabolic Physiology at the Faculty of Physical Education of the Federal University of Uberlândia, Brazil, and was approved by the Ethics Committee for Studies in Humans (CAEE: 47905521.9.0000.5152).

### 2.2. Variables and Data Collection

Multimorbidity was assessed by the presence of ≥2 chronic diseases, with a medical diagnosis documented in their medical record, and with or without drug treatment, in the three disease patterns suggested by studies of Schäfer et al. [19,20]: cardiometabolic disease (diabetes, dyslipidemias, obesity, heart disease—heart attack, angina, heart failure, or others—brain stroke, asthma or asthmatic bronchitis, cancer, chronic kidney disease, and thyroid disorders); musculoskeletal (arthrosis, arthritis or rheumatism, osteoporosis, osteopenia, vertebral, or discotheque problems); and neuropsychological disorders (depression, general anxiety disorder, bipolar and mood disorders, Alzheimer’s disease, epilepsy, and schizophrenia).

The independent variables were as follows: age group (40 to 49 years; 50 to 59 years; 60 to 69 years; and 70 above); menopausal status (pre- or postmenopausal); Body Mass Index (BMI) (Eutrophic until 24.9 kg/m^2^; overweight: 25–29.9 kg/m^2^; obesity I: 30–34.9 kg/m^2^; and obesity II: above ≥35 kg/m^2^); and pattern of disease associated with HT (cardiometabolic, neuropsychological, and musculoskeletal).

### 2.3. Statistical Analysis

To verify the association between the outcome and the exposure variables, the binary logistic regression was used, carried out in the Stata 14.0 software, and a *p* < 0.05 was adopted. The outcome was the presence of multimorbidity and disease patterns (cardiometabolic, neuropsychological, and musculoskeletal). Exposure factors were considered, such as age group, BMI classification, and menopause status. We conducted an unadjusted analysis to explore the relationship between each individual independent variable and the outcome of multimorbidity, without considering any other influencing factors. Additionally, we performed an adjusted analysis, where we simultaneously modeled the relationships between multiple independent variables and the outcome of multimorbidity. This approach enabled us to observe the effect of each variable while controlling for the others.

The sample size was complete according to the formula presented by Tabachnick (2019) [21], which takes into account the number of explanatory variables to be included in the model. N = 50 + 8 m (m is the number of explanatory variables); given that m = 3, a minimum of 74 women will be recruited in this study.

## 3. Results

We analyzed 1003 forms of climacteric women over 40 years old, with a mean age of 63 ± 10 years; among postmenopausal women, the mean age of menopause was 49 years. The sample information is described in Table 1. The frequency of multimorbidity associated with HT was 84%, with the cardiometabolic pattern being the most associated with HT in 86%, and polypharmacy (four or more medications) present in 64% of the sample. Dyslipidemia was the most frequently reported disease (50%), followed by type 2 diabetes (37%) and thyroid disorders (18%).

In Table 2, menopause status and age were associated (*p* < 0.05) with the occurrence of multimorbidity, but without association with BMI. However, in the adjusted analysis, the association between menopause status and the age group of 70 years and above was maintained, with postmenopausal women having a 2.17 times greater chance of being affected by multimorbidity and 5.66 times for women aged 70 years or above.

Table 3 shows the adjusted results of the logistic regression in which the outcome is the association of arterial HT with the analyzed disease pattern (cardiometabolic, neuropsychological, and musculoskeletal). Factors have been shown to be associated in different ways according to the pattern of disease associated with HT.

The risk factors were associated with disease patterns in different ways. Menopausal status was only associated with musculoskeletal diseases, with postmenopausal women having a 2.41 times greater risk of developing these conditions compared to premenopausal women. The age group factor showed an association only with cardiometabolic diseases, in which an increasing risk was demonstrated with each subsequent decade, ranging from 2.10 for women aged 50 to 59 years, 4.28 for 60 to 69 years, and 6.15 for women over 70 years. BMI, on the other hand, proved to be a risk factor for both the cardiometabolic and musculoskeletal disease patterns associated with HT. For cardiometabolic diseases, the risk was 2.30 for obesity grade I and 1.71 for obesity grade II and above. For musculoskeletal diseases, the risk was 1.92 for obesity grade I and 2.34 for obesity grade II and above.

## 4. Discussion

This study investigated the risk factors for multimorbidity and disease patterns in women with HT. In the adjusted analysis, we found an association between multimorbidity, menopause status and age group. In terms of disease patterns, the age group and BMI showed an association with the pattern of cardiometabolic diseases, while for musculoskeletal diseases, the status of menopause, age group, and BMI were considered risk factors. There were no associations of risk factors with the pattern of neuropsychological illness.

In our study, we found a frequency of multimorbidity in 84% of the sample of women with HT over 40 years old. According to a study on the Brazilian population [6], widowed, retired, less educated women with poor self-rated health had a higher prevalence of multimorbidity. Women over 60 years with two or more morbidities reported more hypertension (92.0%) than men did in the same age group (79.5%), especially older and urban residents [6]. In addition, a sex difference was exhibited in the disease pattern. In women, heart disease was the strongest associated factor while in men, pulmonary disease had the highest factor loading [6].

The increased incidence of cardiovascular disease in women, especially climacteric, is related to changes in the concentrations of the hormone estrogen, which plays an important role in cardiovascular control by reducing vascular resistance through the modulation of nitric oxide (NO), the main relaxation factor of the endothelium [22], as well as increasing the synthesis of prostacyclin, an important vasodilator inhibiting the synthesis of vasoconstrictors, such as bradykinin [23]. And in the postmenopausal period, with reduced cardioprotective function, there is an increase in sympathetic activity and an increase in adrenergic vasoconstrictor responsiveness [24]. In our study, it was possible to verify the association of the menopausal status factor with multimorbidity, in which postmenopausal women were 2.17 times more likely to develop ≥2 when compared to premenopausal women.

In addition to the menopause status factor, older age groups (70 years and above) were associated with multimorbidity, being 2.85 times more likely to have ≥2 diseases. The aging process itself results in impairment of the functioning of cardiometabolic systems [25], but other factors also seem to have a considerable influence, such as socioeconomic factors, absence of healthy lifestyle habits, and, for women, the time of menopause [26]. Evidence suggests [27] a positive correlation between high BP (systolic and diastolic BP) and the time since menopause, with the association of these pressure changes being related to the time after menopause and not to the women’s age, per se. Therefore, a longer absence of female gonadal steroids represents an important factor contributing to increased BP in older women [27]. Earlier natural or surgical menopause has also been linked to a higher risk of cardiovascular disease [28].

In our study, the cardiometabolic disease pattern was most strongly associated with HT (86%), followed by chronic conditions such as dyslipidemia (50%), type 2 diabetes (37%), and thyroid disorders (18%). The similarity between risk factors and pathophysiological mechanisms makes individuals with HT, especially those with long-term illness, more likely to develop a second chronic condition [29]. Mechanisms such as inappropriate activation of the renin–angiotensin–aldosterone system, systemic inflammation, inefficient insulin vasodilation, increased activation of the sympathetic nervous system, and oxidative stress secondary to the excessive production of reactive oxygen species (ROS) are shared between the diseases of high BP and type 2 diabetes, for example [30].

Postmenopausal women were 2.41 times more likely to have a pattern of musculoskeletal disease associated with HT. The risk increased with higher BMI classifications: from 1.92 times for Grade 1 obesity (≥30 30 kg/m^2^) to 2.34 times for a BMI of ≥35 30 kg/m^2^.

Because it has receptors located in the tissues of the joint components, estrogen plays a relevant role in maintaining homeostasis and protecting the development of osteoarthritis and osteoporosis, regulating the activity and expression of the main signaling molecules in several different pathways [31]. Even in the absence of injury, the properties of joint tissues undergo adaptive changes with age, genetics, sex (hormones), and environment (biological, mechanical), and, added to the hypoestrogenism resulting from menopause, excess body fat results in the presence of two factors that will compromise the articular and bone tissues. Obesity generates mechanical stress represented by an abnormal load on the joints, resulting in an inflammatory state and other degenerations such as cartilage erosion, microcapillary rupture, and microstructural changes [32].

In our research, we found that 84% of the women in the Hiperdia program experienced multimorbidity, and nearly 64% were using at least four different medications. Previous studies have indicated that polypharmacy can lead to a greater disease burden and issues with treatment adherence [33]. Previous studies indicate that women, especially as they age, often take on more caregiving responsibilities [9] and have to receive more informal care than men [34]. Conversely, men tend to demonstrate higher self-care adherence and experience a lower disease burden [35]. This situation makes postmenopausal women with hypertension a particularly vulnerable group, as they are at an elevated risk for developing multimorbidity. Future research should prioritize an examination of the cultural and social determinants that influence the prevalence of multimorbidity in postmenopausal women.

Despite the concept of multimorbidity being used worldwide in studies, its measurement still faces difficulties in terms of standardization. The use of clusters or disease counting is the method used to structure and study this condition, but as they are still divergent, it has limited comparisons between studies or even in the prevalence between different populations [2,36,37]. However, an analysis by disease pattern provides insight not only into the number of coexisting diseases but also into which specific disease combinations and population characteristics contribute to them. As suggested by Schäfer et al., [20] using clusters/patterns makes it possible to capture a comprehensive picture of the disease patterns in a group of patients or sex differences, reducing the complexity due to the heterogeneity of multimorbidity since some diseases are responsible for overlapping clusters. Assisting in the development of future guidelines and public policies will be instrumental in guiding treatment and avoiding care duplication, thereby enhancing treatment adherence and improving disease management. [2,38].

Some limitations of the study must be addressed. Although all diseases were confirmed through medical diagnosis, we did not have access to the duration of the disease, and, in the case of women with multimorbidity, we cannot determine whether arterial HT was the primary condition or secondary to other conditions. In addition, information such as physical activity level and smoking was not analyzed and could help with additional understanding of the risk factors, as well as the contextual determinants on the socioeconomic level, such as education level and the level and access to self-care, which could produce important complementary associations about multimorbidity. Another important consideration is that the cross-sectional design of this study precludes the establishment of causal relationships between the variables analyzed.

## 5. Conclusions

The findings indicate that women with arterial hypertension are at a significantly higher risk of developing multimorbidity during the postmenopausal period, particularly after the age of 60. This risk is further exacerbated by obesity and the presence of cardiometabolic comorbidities, suggesting a synergistic interplay between hypertension, metabolic dysfunction, and aging. These findings suggest the need for integrated clinical strategies that target weight management, cardiometabolic health, and the prevention of early multimorbidity in this vulnerable population.

## Figures and Tables

**Table 1 ijerph-22-01360-t001:** Sample characteristics (*n* = 1003).

Age Group	%	*n*
40–49	14	142
50–59	27	270
60–69	29	294
≥70 years	30	297
Menopause Status		
No	17	167
Yes	83	814
Medication		
0 to 3 Medications	36	357
4 to 6 Medications	41	414
7 and Above	23	232
Body Mass Index		
Normal	18	156
Overweight	34	283
Obesity I	26	217
Obesity II and Above	22	188
Number of Diseases		
Only HT	16	159
2 or More Diseases	84	844
Multimorbidity Pattern (*n* = 844)		
HT + Cardiometabolic	86	728
HT + Neuropsychological	38	318
HT + Musculoskeletal	24	198
HT + 3 Patterns	5	43
Most Frequent Illnesses		
Dyslipidemias	50	501
Type 2 Diabetes	37	370
Thyroid Disorders	18	180
General Anxiety Disorder	14	139
Depression	12	123
Arthrosis/Arthritis	10	101

HT: hypertension.

**Table 2 ijerph-22-01360-t002:** Logistic regression using multimorbidity (≥2 diseases) as the outcome and age group and BMI classification variables as exposure variables (*n* = 844).

Multimorbidity	Unadjusted Analysis		Adjusted Analysis ^a^	
Variables	OR (CI-95%)	*p*	OR (CI-95%)	*p*
Menopause Status				
No	1		1	
Yes	3.85 (2.62–5.67)	<0.01	2.17 (1.05–4.48)	0.04
Age Group				
40–49 years	1		1	
50–59 years	2.39 (1.51–3.79)	<0.01	1.63 (0.79–3.39)	0.19
60–69 years	4.15 (2.53–6.83)	<0.01	1.92 (0.81–4.56)	0.14
≥70 years	5.66 (3.34–9.62)	<0.01	2.85 (1.16–6.99)	0.02
BMI (kg/m^2^)				
Eutrophy	1		1	
Overweight	1.29 (0.75–2.22)	0.92	1.41 (0.80–2.48)	0.29
Obesity I	1.16 (0.66–2.03)	0.61	1.48 (0.81–2.69)	0.20
Obesity II and above	1.14 (0.64–2.04)	0.65	1.61 (0.87–2.99)	0.13

OR: odds ratio; CI: confidence interval; and BMI: Body Mass Index. ^a^ Adjusted for menopause status, age group and BMI.

**Table 3 ijerph-22-01360-t003:** Logistic regression using cardiometabolic, neuropsychological, and musculoskeletal disease patterns as an outcome (*n* = 844).

Variables	Cardiometabolic	*p*	Neuropsychological	*p*	Musculoskeletal	*p*
	OR (CI-95%)		OR (CI-95%)		OR (CI-95%)	
Menopause Status						
No	1		1		1	
Yes	1.43 (0.77–2.66)	0.261	0.98 (0.52–1.85)	0.06	2.41 (1.05–5.51)	0.04
Age Group						
40–49 years	1		1		1	
50–59 years	2.10 (1.10–3.98)	0.02	0.91 (0.47–1.77)	0.79	1.25 (0.52–3.00)	0.61
60–69 years	4.28 (2.05–8.97)	<0.01	0.72 (0.35–1.51)	0.38	0.86 (0.34–2.19)	0.75
>70 years	6.15 (2.86–13.24)	<0.01	0.91 (0.43–1.90)	0.80	1.08 (0.42–2.75)	0.87
Body Mass Index						
Eutrophy	1		1		1	
Overweight	1.41 (0.88–2.26)	0.15	0.93 (0.61–1.42)	0.76	1.59 (0.91–2.79)	0.10
Obesity I	2.30 (1.36–3.89)	<0.01	0.63 (0.39–1.00)	0.05	1.92 (1.08–3.43)	0.03
>Obesity II	1.71 (1.02–2.88)	0.04	1.04 (0.66–1.65)	0.86	2.34 (1.30–4.24)	<0.01

OR: odds ratio; CI: confidence interval.

## Data Availability

The data that supports the findings of this study are available from the corresponding author, Puga, G.M., upon reasonable request.

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
