# Peer review of "Prevalence, Risk Factors, and Multimorbidity Patterns in Climacteric Women with Hypertension"

_ijerph, 2025, doi:10.3390/ijerph22091360_

Round 1
Reviewer 1 Report
Comments and Suggestions for Authors
General Comment:
The manuscript addresses a relevant public health issue—the patterns and risk factors associated with multimorbidity among climacteric women with hypertension. The statistical analysis is solid, and the data are appropriately interpreted. However, several minor revisions are necessary to improve the clarity, language quality, and consistency of the text.
Specific Comments:
Abstract:
Line 15: Change “is still poorly understood” to “remains poorly understood” for better flow.
Line 20–21: “Age >70” should be revised as “age ≥70 years” for consistency with the rest of the paper.
Line 22: Replace “particularly in women aged ≥50” with “especially among women aged ≥50 years.”
Line 24: “Linked to” is vague; consider “associated with.”
Line 26: “Notably, 86% of hypertensive women had multimorbidity” is better placed earlier in the paragraph for impact.
Introduction:
Line 42: Typo in “medica-tions” → “medications.”
Lines 55–57: Consider restructuring for clarity: “Although BP levels increase with age in both sexes, the increase per decade is greater in women (8.1 mmHg) than in men (4.7 mmHg).”
Lines 59–61: Sentence could be rephrased for clarity: “Due to aging, hypoestrogenism, and lifestyle factors, climacteric women are increasingly presenting multiple chronic diseases.”
Methods:
Line 71: Consider specifying the year or period of data collection.
Line 88–89: Clarify BMI groups with hyphenation or semicolons for readability.
Lines 79–85: The definition of disease patterns is helpful, but some conditions (e.g., asthma and cancer) under "cardiometabolic" are not clearly justified there—consider further clarification or citing precedent.
Results:
Tables 2 and 3: Ensure formatting is consistent and clear. In Table 2, headings like "Unadjusted Analysis" and "Adjusted Analysis" should be clearly delineated.
Line 109: Consider rewriting “most frequent disease reported” as “most frequently reported disease.”
Line 114–116: The phrase “people aged 70 years or above” is vague—consider replacing “people” with “women.”
Discussion:
Line 143–144: “Women had a higher prevalence (96%) of arterial HT than men (79.5%)” seems unusually high—please clarify if this refers to hypertensive individuals only.
Line 170: “Exposure has also been demonstrated...” – revise for clarity. Suggestion: “Earlier natural or surgical menopause has also been linked to a higher risk of cardiovascular disease.”
Line 195: Typo in “microstructure change” → consider “microstructural changes.”
Lines 207–213: Limitations are appropriate. However, you might add that the cross-sectional design prevents causal inference.
Conclusions:
Line 215: “Face a significantly higher risk of developing multimorbidity” → “are at significantly higher risk of developing multimorbidity.”
Line 220: Consider softening “These results underscore the need…” to “These findings suggest the need for…”
Language & Style:
Improve consistency in terminology: use “hypertension” or “HT” uniformly.
Prefer “postmenopausal women” over “women in postmenopause.”
Avoid contractions (e.g., “can’t”) in scientific writing.
Some subject–verb agreement and article usage (e.g., “the pattern,” “a greater risk”) need polishing throughout.
Recommendation: Minor Revision
The manuscript is scientifically sound, relevant, and well-structured. However, it requires minor revisions regarding language, clarity, and consistency in formatting and terminology. Once these issues are addressed, the paper will be suitable for publication.
Author Response
General Comment:
The manuscript addresses a relevant public health issue—the patterns and risk factors associated with multimorbidity among climacteric women with hypertension. The statistical analysis is solid, and the data are appropriately interpreted. However, several minor revisions are necessary to improve the clarity, language quality, and consistency of the text.
Specific Comments:
Abstract:
Line 15: Change “is still poorly understood” to “remains poorly understood” for better flow.
Response: Thank you for the suggestion. We have made the change in the manuscript.
Line 20–21: “Age >70” should be revised as “age ≥70 years” for consistency with the rest of the paper.
Response: Thank you for the suggestion. We have made the change in the manuscript.
Line 22: Replace “particularly in women aged ≥50” with “especially among women aged ≥50 years.”
Response: Thank you for the suggestion. We have made the change in the manuscript.
Line 24: “Linked to” is vague; consider “associated with.”
Response: Thank you for the suggestion. We have made the change in the manuscript.
Line 26: “Notably, 86% of hypertensive women had multimorbidity” is better placed earlier in the paragraph for impact.
Response: Thank you for the suggestion. We have made the change in the manuscript.
Introduction:
Line 42: Typo in “medica-tions” → “medications.”
Response: Thank you for pointing this out. The journal’s template automatically formats the text, so we were unable to make this adjustment manually
Lines 55–57: Consider restructuring for clarity: “Although BP levels increase with age in both sexes, the increase per decade is greater in women (8.1 mmHg) than in men (4.7 mmHg).”
Response: Thank you for the suggestion. We have restructured the sentence accordingly.
Lines 59–61: Sentence could be rephrased for clarity: “Due to aging, hypoestrogenism, and lifestyle factors, climacteric women are increasingly presenting multiple chronic diseases.”
Response: Thank you for the suggestion. We have rephrased the sentence accordingly.
Methods:
Line 71: Consider specifying the year or period of data collection.
Response: Thank you for the suggestion. Data were collected between October 11 and December 22, 2021. This information has been added to the manuscript.
Line 88–89: Clarify BMI groups with hyphenation or semicolons for readability.
Response: Thank you for the suggestion. We have revised the text to improve readability.
Lines 79–85: The definition of disease patterns is helpful, but some conditions (e.g., asthma and cancer) under "cardiometabolic" are not clearly justified there—consider further clarification or citing precedent.
Response: In order to capture a comprehensive picture of the disease patterns in individual patients, we organized the list of chronic conditions based on the three clusters used by Schäfer et al 2010 [1] and suggested by Schäfer et al 2014 [2] to reduce complexity due to heterogeneity of multimorbidity since some diseases are responsible for overlapping clusters.
We have added your helpful commentary to our methods session and added a section to the discussion, which has highlighted the references mentioned.
Results:
Tables 2 and 3: Ensure formatting is consistent and clear. In Table 2, headings like "Unadjusted Analysis" and "Adjusted Analysis" should be clearly delineated.
Response: Thank you for your suggestion. The table formatting has been standardized, and a legend has been added to both tables to describe the adjustments made. Furthermore, a paragraph has been included in the statistical analysis section.
“We conducted an unadjusted analysis to explore the relationship between each individual independent variable and the outcome of multimorbidity, without taking into account any other influencing factors. Additionally, we performed an adjusted analysis where we simultaneously modelled the relationships between multiple independent variables and the outcome of multimorbidity. This approach enabled us to observe the effect of each variable while controlling for the others.”
Line 109: Consider rewriting “most frequent disease reported” as “most frequently reported disease.”
Response: Thank you for the suggestion. We have revised the sentence accordingly.
Line 114–116: The phrase “people aged 70 years or above” is vague—consider replacing “people” with “women.”
Response: Thank you for the suggestion. We have made the replacement.
Discussion:
Line 143–144: “Women had a higher prevalence (96%) of arterial HT than men (79.5%)” seems unusually high—please clarify if this refers to hypertensive individuals only.
Response: We have clarified this sentence as follows:
“According to a study with the Brazilian population, women exhibited a higher prevalence of multimorbidity, with higher risk in widowers, with lower education, retired and who had the worst perceptions of health, with the highest risk. Compared to men, women over 60 years with two or more morbidities reported more hypertension (92.0%) than men did in the same age group (79.5%) [3].”
Line 170: “Exposure has also been demonstrated...” – revise for clarity. Suggestion: “Earlier natural or surgical menopause has also been linked to a higher risk of cardiovascular disease.”
Response: Thank you for the suggestion. We have revised the sentence accordingly.
Line 195: Typo in “microstructure change” → consider “microstructural changes.”
Response: Thank you for the suggestion. We have made the replacement.
Lines 207–213: Limitations are appropriate. However, you might add that the cross-sectional design prevents causal inference.
Response: Thank you for the suggestion. We have added this limitation to the manuscript.
Conclusions:
Line 215: “Face a significantly higher risk of developing multimorbidity” → “are at significantly higher risk of developing multimorbidity.”
Response: Thank you for the suggestion. We have made the replacement.
Line 220: Consider softening “These results underscore the need…” to “These findings suggest the need for…”
Response: Thank you for the suggestion. We have made the change.
Language & Style: Improve consistency in terminology: use “hypertension” or “HT” uniformly.
Response: Thank you for the suggestion. We have standardized the acronym throughout the manuscript.
Prefer “postmenopausal women” over “women in postmenopause.”
Response: Thank you for the suggestion. We have made the change.
Avoid contractions (e.g., “can’t”) in scientific writing.
Response: Thank you for the suggestion. We have removed contractions throughout the manuscript.
Some subject–verb agreement and article usage (e.g., “the pattern,” “a greater risk”) need polishing throughout.
Response: We have reviewed the entire manuscript and made the necessary adjustments.
References:
- Schäfer I, von Leitner EC, Schön G, et al. Multimorbidity Patterns in the Elderly: A New Approach of Disease Clustering Identifies Complex Interrelations between Chronic Conditions. Ross JS, ed. PLoS One. 2010;5(12):e15941. doi:10.1371/journal.pone.0015941
- Schäfer I, Kaduszkiewicz H, Wagner HO, Schön G, Scherer M, van den Bussche H. Reducing complexity: a visualisation of multimorbidity by combining disease clusters and triads. BMC Public Health. 2014;14(1):1285. doi:10.1186/1471-2458-14-1285
- Araujo MEA, Silva MT, Galvao TF, Nunes BP, Pereira MG. Prevalence and patterns of multimorbidity in Amazon Region of Brazil and associated determinants: a cross-sectional study. BMJ Open. 2018;8(11):e023398. doi:10.1136/bmjopen-2018-023398
- Secretaria de Comunicação Social. Unidades Básicas de Saúde. Ações e programas do Governo Federal. 2024. https://www.gov.br/secom/pt-br/acesso-a-informacao/comunicabr/lista-de-acoes-e-programas/unidades-basicas-de-saude-do-governo-federal
- Rudnicka E, Napierała P, Podfigurna A, Męczekalski B, Smolarczyk R, Grymowicz M. The World Health Organization (WHO) approach to healthy ageing. Maturitas. 2020;139:6-11. doi:10.1016/j.maturitas.2020.05.018
Reviewer 2 Report
Comments and Suggestions for Authors
Dear authors: The method chosen and the source of the data are unclear. The objective of this study is to identify patterns of associated diseases and the risk factors specific to each pattern. Postmenopausal women were 2.41 times more likely to have the pattern of musculo-skeletal disease associated with arterial hypertension, as well as increasing the BMI classification. Grade 1 obesity (≥30 kg/m2) increases the risk from 1.92 to 2.34 times more likely with a BMI of ≥35 kg/m2. In relation to what? In general, no response was observed to the objective set forth in the article, since it is not possible to clearly associate the specific risks of each pattern, although they are possible or diffuse, as stated at the end in the limitations section. Regarding the morbidity and mortality mentioned in the title, this is also unclear.The bibliography should be updated. Regards
Author Response
The method chosen and the source of the data are unclear. The objective of this study is to identify patterns of associated diseases and the risk factors specific to each pattern. Postmenopausal women were 2.41 times more likely to have the pattern of Musculo-skeletal disease associated with arterial hypertension, as well as increasing the BMI classification. Grade 1 obesity (≥30 kg/m2) increases the risk from 1.92 to 2.34 times more likely with a BMI of ≥35 kg/m2. In relation to what? In general, no response was observed to the objective set forth in the article, since it is not possible to clearly associate the specific risks of each pattern, although they are possible or diffuse, as stated at the end in the limitations section. Regarding the morbidity and mortality mentioned in the title, this is also unclear. The bibliography should be updated. Regards.
Response: Thanks for the comments. Basic Health Units (UBS) are primary healthcare centers where Family Health teams carry out a range of health actions. They represent the main entry point to the Brazilian Unified Health System (SUS), addressing individual and collective health needs. UBSs serve the general population, offering primary health care, including prevention, diagnosis, and treatment. They are an integral part of the SUS, regulated by federal laws and regulations that guide their structure and operation [4]. We added this information to the text to make it clearer.
The data for this study was collected from a system between October 11 and December 22, 2021, from an analysis of the medical records of participants in the Hyperdia Program, a SUS program that monitors people with hypertension and/or diabetes. The program functions as a primary care tool, offering continuous follow-up and monitoring of the blood pressure and glucose levels of these patients. We added this information to the text to make it clearer.
From the collected data, we performed a cross-sectional analysis to understand the patterns of associated diseases and the specific risk factors for each pattern in climacteric women with hypertension. A cross-sectional study is a type of observational study that analyzes data from a population at a single point in time, simultaneously assessing exposure and outcome. It is often used in public health, and in the case of the present study, it was used to determine the prevalence of diseases, risk factors, and multimorbidity patterns in climacteric women.
The disadvantage of this chosen method, as you suggested in your review and as we included in the study's limitation section, is that the cross-sectional analysis prevents the establishment of causal relationships between the variables analyzed. In contrast, this cross-sectional analysis carried out in UBSs provided important answers for public health, as it allowed us to identify that women with arterial hypertension are at a significantly higher risk of developing multimorbidity during the postmenopausal period, particularly after the age of 60. This risk is further exacerbated by obesity and the presence of cardiometabolic comorbidities, suggesting a synergistic interplay between HT, metabolic dysfunction, and aging. These findings suggest the need for integrated clinical strategies aimed at weight management, cardiometabolic health, and early multimorbidity prevention in this vulnerable population.
In the present study, our analysis found that menopausal status was associated only with musculoskeletal diseases, with postmenopausal women having a 2.41 times greater risk of developing these conditions compared to premenopausal women. The age group factor showed an association only with cardiometabolic diseases, with the risk increasing progressively with each decade: 2.10 for women aged 50 to 59, 4.28 for 60 to 69, and 6.15 for women aged 70 or older. BMI, on the other hand, was a risk factor for both cardiometabolic and musculoskeletal disease patterns. For cardiometabolic diseases, the risk was 2.30 for obesity grade I and 1.71 for obesity grade II or higher. For musculoskeletal diseases, the risk was 1.92 for obesity grade I and 2.34 for obesity grade II or higher. We have rewritten this paragraph in the manuscript to make the explanation clearer.
Regarding the term "Multimorbidity" in the title "Prevalence, Risk Factors and Multimorbidity Patterns in Climacteric Women with Hypertension", it is defined, according to the WHO [5], as the co-occurrence of two or more chronic medical conditions in a person, which includes physical and mental health conditions. The conceptualization of the term is found in the first sentence of the introduction. Furthermore, we did not identify the term "mortality" in the title, so we cannot clarify the mentioned doubt. As requested, we have updated the article's bibliography. Regards.
References:
- Schäfer I, von Leitner EC, Schön G, et al. Multimorbidity Patterns in the Elderly: A New Approach of Disease Clustering Identifies Complex Interrelations between Chronic Conditions. Ross JS, ed. PLoS One. 2010;5(12):e15941. doi:10.1371/journal.pone.0015941
- Schäfer I, Kaduszkiewicz H, Wagner HO, Schön G, Scherer M, van den Bussche H. Reducing complexity: a visualisation of multimorbidity by combining disease clusters and triads. BMC Public Health. 2014;14(1):1285. doi:10.1186/1471-2458-14-1285
- Araujo MEA, Silva MT, Galvao TF, Nunes BP, Pereira MG. Prevalence and patterns of multimorbidity in Amazon Region of Brazil and associated determinants: a cross-sectional study. BMJ Open. 2018;8(11):e023398. doi:10.1136/bmjopen-2018-023398
- Secretaria de Comunicação Social. Unidades Básicas de Saúde. Ações e programas do Governo Federal. 2024. https://www.gov.br/secom/pt-br/acesso-a-informacao/comunicabr/lista-de-acoes-e-programas/unidades-basicas-de-saude-do-governo-federal
- Rudnicka E, Napierała P, Podfigurna A, Męczekalski B, Smolarczyk R, Grymowicz M. The World Health Organization (WHO) approach to healthy ageing. Maturitas. 2020;139:6-11. doi:10.1016/j.maturitas.2020.05.018
Reviewer 3 Report
Comments and Suggestions for Authors
Dear authors,
Thank you for submitting your manuscript. Overall, I would like to congratulate you on your work, although I must offer several comments that I hope will help you improve your research. Please find my observations below:
Introduction:
The introduction appropriately addresses the definition of multimorbidity, its impact on the healthcare system, and justifies the focus on climacteric women. However, it could be improved by including a conceptual framework or a perspective that adds originality to the study. As the topic has already been widely addressed in previous studies, it would be helpful to explain what new insights this research offers or to further justify the relevance and publication of the manuscript.
Methods:
The cross-sectional design is appropriate for estimating prevalence and associations between risk factors and disease patterns in a broad population. The methodological justification is clear and consistent with the study objective. The methods section includes inclusion criteria, independent variables, and the statistical analysis strategy. However, this section could be expanded to explore additional variations. For example, it would be relevant to analyse associations by subgroups combining characteristics such as menopausal status across different age ranges. These distinctions might help explain subsequent associations. That said, the methodology is sufficiently described and allows replication.
Results:
The results are understandable, well-organised, and supported by specific prevalence data and odds ratios. Key associations are well interpreted. The tabular presentation in the original manuscript is appropriate.
Discussion:
The discussion is adequate, although it could be strengthened by including more references, particularly recent ones, as some are nearly 20 years old. Moreover, the discussion shows that a deductive methodology (logic-based reasoning) could lead to similar conclusions based on existing knowledge about pathophysiology and care. For instance, postmenopausal women with obesity tend to experience more health problems, which is related to factors such as ageing, hormonal changes, and accumulated care burdens over the years. Would differences be expected if women had engaged in self-care? Could there be variations depending on socioeconomic resources or the level of informal care demands in the household? These factors are key to understanding the differences beyond what can be inferred physiologically.
Conclusions:
The conclusions are well grounded in the results and appropriately highlight the identified risk factors.
References:
The references in the introduction could be updated and expanded to include literature on gender differences in multimorbidity and international comparative studies, which would better contextualise the study’s relevance and originality. Likewise, the references in the discussion are limited, especially for a topic that is widely studied and documented.
I hope these comments are helpful.
Thank you very much.
Author Response
Dear authors,
Thank you for submitting your manuscript. Overall, I would like to congratulate you on your work, although I must offer several comments that I hope will help you improve your research. Please find my observations below:
Introduction:
The introduction appropriately addresses the definition of multimorbidity, its impact on the healthcare system, and justifies the focus on climacteric women. However, it could be improved by including a conceptual framework or a perspective that adds originality to the study. As the topic has already been widely addressed in previous studies, it would be helpful to explain what new insights this research offers or to further justify the relevance and publication of the manuscript.
Response: We appreciate your suggestions. We have added two paragraphs at the end of the Introduction to present the new insights our research offers and to further justify the relevance and originality of publishing the manuscript.
Methods:
The cross-sectional design is appropriate for estimating prevalence and associations between risk factors and disease patterns in a broad population. The methodological justification is clear and consistent with the study objective. The methods section includes inclusion criteria, independent variables, and the statistical analysis strategy. However, this section could be expanded to explore additional variations. For example, it would be relevant to analyse associations by subgroups combining characteristics such as menopausal status across different age ranges. These distinctions might help explain subsequent associations. That said, the methodology is sufficiently described and allows replication.
Response: We thank the reviewer for the comment and agree that the analysis of associations in subgroups, combining menopausal status and different age ranges, would be of great relevance and could provide additional insights into the topic. However, the cross-sectional design of our study, while appropriate for the objective of estimating prevalence and associations in a single population at one point in time, was not optimized for conducting complex interaction analyses between variables. A more detailed subgroup analysis could be limited by the sample size of some subdivisions, which would reduce the statistical power and robustness of the results. We believe that the current methodology, which analyzes risk factors separately, was sufficient to address the primary objectives of the study, which were to identify the patterns of associated diseases and the specific risk factors for each pattern.
Results:
The results are understandable, well-organised, and supported by specific prevalence data and odds ratios. Key associations are well interpreted. The tabular presentation in the original manuscript is appropriate.
Response: We appreciate the reviewer’s positive feedback on the presentation and clarity of our results.
Discussion:
The discussion is adequate, although it could be strengthened by including more references, particularly recent ones, as some are nearly 20 years old. Moreover, the discussion shows that a deductive methodology (logic-based reasoning) could lead to similar conclusions based on existing knowledge about pathophysiology and care. For instance, postmenopausal women with obesity tend to experience more health problems, which is related to factors such as ageing, hormonal changes, and accumulated care burdens over the years. Would differences be expected if women had engaged in self-care? Could there be variations depending on socioeconomic resources or the level of informal care demands in the household? These factors are key to understanding the differences beyond what can be inferred physiologically.
Response: Thank you for your suggestions. We have incorporated more recent references as recommended and added an extra paragraph to the discussion highlighting the significance of factors like self-care and treatment.
Conclusions:
The conclusions are well grounded in the results and appropriately highlight the identified risk factors.
Response: We appreciate the reviewer’s positive feedback on the strength and alignment of our conclusions.
References:
The references in the introduction could be updated and expanded to include literature on gender differences in multimorbidity and international comparative studies, which would better contextualise the study’s relevance and originality. Likewise, the references in the discussion are limited, especially for a topic that is widely studied and documented.
Response: We have included a paragraph in both the introduction and discussion sections that addresses the differences between the sexes.
References:
- Schäfer I, von Leitner EC, Schön G, et al. Multimorbidity Patterns in the Elderly: A New Approach of Disease Clustering Identifies Complex Interrelations between Chronic Conditions. Ross JS, ed. PLoS One. 2010;5(12):e15941. doi:10.1371/journal.pone.0015941
- Schäfer I, Kaduszkiewicz H, Wagner HO, Schön G, Scherer M, van den Bussche H. Reducing complexity: a visualisation of multimorbidity by combining disease clusters and triads. BMC Public Health. 2014;14(1):1285. doi:10.1186/1471-2458-14-1285
- Araujo MEA, Silva MT, Galvao TF, Nunes BP, Pereira MG. Prevalence and patterns of multimorbidity in Amazon Region of Brazil and associated determinants: a cross-sectional study. BMJ Open. 2018;8(11):e023398. doi:10.1136/bmjopen-2018-023398
- Secretaria de Comunicação Social. Unidades Básicas de Saúde. Ações e programas do Governo Federal. 2024. https://www.gov.br/secom/pt-br/acesso-a-informacao/comunicabr/lista-de-acoes-e-programas/unidades-basicas-de-saude-do-governo-federal
- Rudnicka E, Napierała P, Podfigurna A, Męczekalski B, Smolarczyk R, Grymowicz M. The World Health Organization (WHO) approach to healthy ageing. Maturitas. 2020;139:6-11. doi:10.1016/j.maturitas.2020.05.018